# What, When and How to Measure—Peripheral Biomarkers in Therapy of Huntington’s Disease

**DOI:** 10.3390/ijms22041561

**Published:** 2021-02-04

**Authors:** Lukasz Przybyl, Magdalena Wozna-Wysocka, Emilia Kozlowska, Agnieszka Fiszer

**Affiliations:** 1Laboratory of Mammalian Model Organisms, Institute of Bioorganic Chemistry Polish Academy of Sciences, 61-704 Poznan, Poland; 2Department of Medical Biotechnology, Institute of Bioorganic Chemistry Polish Academy of Sciences, 61-704 Poznan, Poland; mwozna@ibch.poznan.pl (M.W.-W.); emiliak@ibch.poznan.pl (E.K.)

**Keywords:** Huntington’s disease, biomarkers, periphery, immune, therapy, microbiome, miRNA, blood

## Abstract

Among the main challenges in further advancing therapeutic strategies for Huntington’s disease (HD) is the development of biomarkers which must be applied to assess the efficiency of the treatment. HD is a dreadful neurodegenerative disorder which has its source of pathogenesis in the central nervous system (CNS) but is reflected by symptoms in the periphery. Visible symptoms include motor deficits and slight changes in peripheral tissues, which can be used as hallmarks for prognosis of the course of HD, e.g., the onset of the disease symptoms. Knowing how the pathology develops in the context of whole organisms is crucial for the development of therapy which would be the most beneficial for patients, as well as for proposing appropriate biomarkers to monitor disease progression and/or efficiency of treatment. We focus here on molecular peripheral biomarkers which could be used as a measurable outcome of potential therapy. We present and discuss a list of wet biomarkers which have been proposed in recent years to measure pre- and postsymptomatic HD. Interestingly, investigation of peripheral biomarkers in HD can unravel new aspects of the disease pathogenesis. This especially refers to inflammatory proteins or specific immune cells which attract scientific attention in neurodegenerative disorders.

## 1. Introduction

Huntington’s disease (HD) is a neurodegenerative disorder characterized by motor and cognitive dysfunction [1]. Concerning its genetic background, HD is one of nine polyglutamine (polyQ) diseases which are caused by CAG repeat expansion in the open reading frame (ORF) region of specific genes. These diseases also include dentatorubral-pallidoluysian atrophy (DRPLA); spinal bulbar muscular atrophy (SBMA); and several spinocerebellar ataxias (SCAs), type 1, 2, 3, 6, 7 and 17 [2]. There is an emerging need to develop an effective therapy for these diseases, because they remain incurable. Symptoms associated with progressive neurodegeneration in the brain usually appear in the fourth or fifth decade of life and lead to death within 10–15 years [3].

In HD, CAG repeat expansion located in the ORF of *HTT* gene leads to the production of huntingtin protein containing abnormally elongated polyQ tract. Normally, *HTT* contains ~15 CAG repeats located in the first exon, whereas HD patients usually have a mutant tract of 40–50 CAG units at one *HTT* allele. The juvenile form of HD is characterized by 60–120 CAG repeats in *HTT*. PolyQ protein toxicity is connected with its gain-of-toxic-function, as well as dysfunction of important cellular interactions [4]. A hallmark of mutant proteins containing polyQ tracts is aggregate formation. Interestingly, *HTT* expression is described as ubiquitous in human tissues with the highest level found in the cerebral cortex [5].

In HD, like for the majority of neurological disorders, the pathology is mainly linked to the central nervous system (CNS) and the brain as an organ where crucial degeneration occurs. HD pathogenesis leads to a massive loss of medium spiny neurons (MSN) in the striatum and loss of neurons in the cortex as the disease progresses [6]. The resulting clinical symptoms of HD include motor deficits, psychiatric disability and cognitive decline [1]. Overall (brain and periphery) expression of mutant huntingtin may contribute to other symptoms, such as weight loss, glucose metabolism, cardiomyopathy, muscle and endocrine dysfunction, in addition to those associated with neurological pathology [7,8]. The peripheral nervous system (PNS) serves as a direct signal communicator between the CNS and peripheral organs, but another type of indirect signals can be passed through the immune system (IS). For a broader group of neurodegenerative diseases, some common changes can be observed due to neuroinflammation [9]. HD patients suffer from chronic inflammation that can be detected in the periphery several years before the disease onset [10], which is described in detail in Section 4.7.

In recent years, a vast body of evidence has appeared showing pathologies happening in peripheral organs, e.g., the heart. Its dysfunction is among the leading causes of HD patient death. Additionally, early HD patients are diagnosed with bradycardia and prolonged intraventricular conduction [11], as the autonomic nervous system and cardiac remodeling are dysfunctional [12,13].

The goal of this review is to shed light on the molecular hallmarks of HD which could be used as prognostic biomarkers or quantitative outcome measures of disease-modifying therapeutic approaches. We briefly present the most advanced therapeutic strategies which aim to down-regulate mutant huntingtin. We mainly focus on the non-invasive and minimally invasive molecular biomarkers which are potentially the most useful for the estimation of treatment efficiency.

## 2. Disease-Modifying Strategies for HD

Potentially the most effective therapeutic approach for patients with HD is the elimination of mutant gene expression. The down-regulation of *HTT* in crucial brain regions is expected to lead to a lack of initiation of pathogenic pathways by mutant protein. Clinical trials of selected disease-modifying strategies are currently underway (Table 1) [14]. The most advanced options are the strategies based on targeting the transcript with specific oligonucleotide-based tools [15]. More than 25 years have passed since the identification of the gene responsible for HD. At the end of 2017, Ionis Pharmaceuticals and Roche completed the key stage in clinical trials for HD using antisense oligonucleotides (ASOs) [16], and further testing is ongoing. The breakthrough observation was the lowering of huntingtin in the patients’ cerebrospinal fluid (CSF), but many challenges are still to be overcome in therapy for HD.

Among the disease-modifying strategies, we can distinguish allele-selective and non-allele-selective approaches [17]. The latter therapy involves the reduction in both normal and mutant allele expression. The allele-selective action, on the other hand, concerns the regulation of the expression of the pathological allele of the gene only. In the case of HD, this would allow the level of the normal *HTT* (wild-type *HTT*, wt *HTT*) allele to be maintained. Normal huntingtin plays an important cellular role, and its down-regulation during long term treatment could cause additional adverse effects [18]. A study published in 2017 showed that a complete reduction in normal huntingtin levels for 9 months in mice led to neurological symptoms, brain atrophy, a broad immune response and iron metabolism dysfunction. Nevertheless, *Htt* elimination in mice was found to be well tolerated in the adult striatum and cortex, which are crucial regions for the delivery of therapeutics in HD [19]. There are no conclusive data for humans, but it is assumed that lowering the total protein level to 50% should be tolerated by patients, although an allele-selective strategy is still considered safer [20]. However, the development of an allele-selective strategy is more difficult than a non-allele-selective one. It requires the design of molecules for regions of a gene or transcript that differ between two alleles, i.e., regions containing different variants of a single nucleotide polymorphism (SNP) or mutation region, i.e., an extended CAG repeat sequence.

Among strategies aiming to selectively target the mutant transcript, RNA interference (RNAi)-based reagents and antisense oligonucleotides (ASOs) are the most prominent tools [15,17]. These approaches require the use of short oligonucleotide (10–22 nt long) complementary to a fragment of specific mRNA and can be designed for transcript degradation or inhibition of translation. In a typical RNAi pathway, one strand of RNA duplex (short interfering RNAs, siRNAs) bind to the complementary sequence in mRNA within the RNA-induced silencing complex (RISC), and subsequently, one of its components, Ago2, cleaves the targeted transcript, which leads to further degradation [21]. RNAi is originally activated by siRNAs but can be also applied in therapeutic approaches using vectors expressing precursors encoding RNA duplexes. Due to their variety in cell- and tissue-specific tropisms, adeno-associated viruses (AAVs) are currently the most attractive delivery vectors for siRNAs, providing long-term expression without genomic integration and relative high safety [22]. More clinical trials using this system were considered after the first AAV-mediated gene therapy was approved by the FDA in 2017 [23].

Another option is the use of chemically modified siRNAs which possess increased stability, as compared to unmodified RNAs, but their periodic delivery is still required. Very promising results were obtained recently with the use of divalent siRNAs (di-siRNAs). In preclinical studies in mice and primates, significant non-allele-selective suppression of *HTT* expression was achieved 6 months after di-siRNA administration [24]. One promising approach for allele-selective therapy for HD is direct oligonucleotide targeting to the mutation site, i.e., the expanded CAG repeat tract. In order to achieve specific silencing of the mutant allele, it was necessary to design atypical siRNA molecules, in some respects similar to miRNA [25,26,27]. They contain specific nucleotide substitutions that result in non-canonical base pairs in interaction with the CAG repeat sequence. Testing of these siRNAs in HD cell models led to the activation of the silencing mechanism preferentially for the mutant allele, due to the binding of multiple RISCs to an extended sequence of repeats [28,29]. Additionally, this strategy is also effective for other polyQ diseases [30,31,32].

ASOs are 16 to 22 base pair single stranded DNA analogues that have been tested in therapeutic approaches since the 1970s. Hence, they have been optimized for a long time, and much is known about their pharmacological properties [33]. Their advantage is resistance to the action of exonuclease, largely due to the presence of an advanced pattern of chemical modifications, ensuring both high efficiency and the absence of side effects related to their metabolism [34,35]. ASOs bind to the pre-mRNA in the cell nucleus, and after recognition of the RNA-DNA complex, the transcript is cleaved by RNase H1 and then degraded [14,15,17,35]. There are also other mechanisms of ASO activity, such as translation inhibition, as well as the disruption of the splicing process [14,35]. As potential therapeutics for neurodegenerative diseases, ASO distributes very well within CSF [17,36].

In potential HD therapy, ASOs were used in both the allele-selective and non-allele-selective approaches for the silencing of *HTT* expression [14,17]. Experiments on mouse HD models showed that using *HTT*-targeting ASOs it is possible to stop the progression of the disease, and additionally the functioning of the nervous system is improved [34]. In addition, it has been shown that lowering total huntingtin levels in the CNS (with maximal 75% reduction achieved) is well tolerated by mice throughout the study, i.e., 3 months [34].

Among the main challenges of *HTT* silencing strategies is to achieve high specificity of action, without the so-called off-target effects. Therapeutic molecules can bind to an unintended region in the genome or transcriptome, which can alter the expression of many genes. Another challenge is the efficient delivery of molecules to brain cells where key pathogenic processes are operating. Due to the blood–brain barrier (BBB), the penetration of most of the molecules is impossible; hence, direct injection of potential therapeutics into the brain is often used in research on rodent models. Special carrier approaches are also being developed that are able to cross the BBB [37]. Delivery of therapeutic molecules to a specific region is considered advantageous due to lower off-target effects. Nevertheless, it may be necessary to target various brain regions and different cell types due to pathogenic interactions in the brain network. Targeting the striatum only for mutant gene down-regulation may show beneficial effects to reverse the disease phenotype in mouse models, but in humans may be insufficient.

## 3. Biomarkers—General Information and Types Applicable for HD

An appropriate biomarker should provide quantitative and reproducible measurement of a parameter which correlates with disease progression. There are three distinguished groups of biomarkers depending on a mode of acquisition. The first is derived from biopsy (invasive) and is a biochemical measure of specific parameters or is histologically evaluated. The second is validated from blood or other fluids (so-called wet biomarkers) and can be represented by the presence or level of some molecules or cell types. The third is typically non-invasive imaging. Reliable and rapid diagnostic tests based on disease-specific biomarkers enable the initial determination of the pathological state, monitoring disease progression and thus the determination of prognostic factors. The clinical significance of such tests is also particularly important in assessing the effectiveness of applied therapies. As preventive treatment is the most beneficial for patients, biomarkers which are related to the premanifest disease state are required. In the case of presymptomatic treatment, prevention or delay of disease onset is expected. Therapeutic intervention at other stages of the disease should lead to a decrease in its progression. This requires very sensitive biomarkers for evaluation if therapy is efficient. Moreover, the combination of several tests might be necessary to address different aspects of the disease progression.

A very important issue in the development of biomarkers is the requirement for standardized and sensitive assays, as well as appropriate statistical rigor in the data analysis. Appropriate methods must be applied in order to obtain reliable results for each patient in the time course and also for a group of patients. This also requires unique standards for bioanalytical laboratories [38]. Another challenge in prospective retrospective biomarker studies is to estimate the power for assessing the prognostic and predictive values of a single biomarker. Appropriate statistical analyses must be used for high quality and statistical rigor in biomarker studies [38].

An attractive option is non-invasive biomarkers which, in the case of polyQ diseases, include different types of brain imaging. This requires advanced equipment with uniform settings to provide the data which can be compared and/or pooled. HD progression is highly correlated with increasing striatal atrophy. This can be assessed using magnetic resonance imaging (MRI) for the measurement of the volume and/or shape of selected brain regions [39]. Importantly, volumetric changes of no more than a few percent can be detected within every year of disease progression [40,41]. Unfortunately, atrophic changes can only be stopped and not reversed by the treatment. Brain imaging oriented at specific metabolites and functional molecules could provide more responsive parameters to therapeutic treatment than structural methods. In this approach, methods such as PET, functional MRI (fMRI) and magnetic resonance spectroscopy (MRS) are used to quantify the neuronal activity and functioning of specific brain regions by measuring specific metabolites. An example is 18F-fluorodeoxyglucose (FDG)-PET analysis used to quantify glucose hypometabolism in the striatum in HD patients [42].

Clinical aspects of biomarkers for HD were addressed in large scale screenings of patients in PREDICT-HD [43] and TRACK-HD programs [44]. These longitudinal observational studies provided data related to premanifest and early-stage HD patients using neuropsychological and imaging measures where detectable changes were reported prior to the predicted time of clinical diagnosis. To evaluate the therapeutic outcome of preclinical studies, a set of specific tests for cognitive and motor measures was used. In the case of patients, rating scales were developed to assess motor and cognitive function, behavioral abnormalities and functional capacity, such as the Unified Huntington’s Disease Rating Scale (UHDRS) (Huntington Study Group 1996). Nevertheless, these types of assessment are mostly applicable to symptomatic patients, and one must consider HD as a progressive disease with a long-lasting presymptomatic stage.

To some extent, biomarkers developed in preclinical experiments with animal models can be transferred to clinical testing with patients and vice versa. One example of a biomarker developed in animal studies and its transfer to patients is a study using HD R6/2 mice in which the treatment with an inhibitor of mitochondrial fragmentation, P110, resulted in the restoration of mitochondrial DNA (mtDNA) levels in plasma [45]. Subsequently, significant changes in mtDNA levels were also found in the plasma of HD patients, as compared to control individuals, indicating the possible use of this biomarker. As another example, one preclinical study used biomarkers for the assessment of functional recovery of the brain in HD mice treated with siRNAs administered with LV vectors [46]. The measurement of succinate dehydrogenase (SDH) activity and cerebral metabolic rate of glucose (CMRGlu) was performed in striatal sections. Corresponding alterations were previously reported in the striatum of HD patients [47].

Biomarkers applied for therapeutic approaches targeting the cause of disease, preferentially presymptomatically, need to be more sensitive than those developed for symptomatic treatments. It is important to emphasize that an applicable biomarker should also possess relatively low variability in measurements in both control and diseased populations. Biomarkers assessed in specific animal models of a disease are analyzed in a relatively uniform genetic background. Due to that fact, one might assume that specific biomarkers may be more variable in patients.

Among the critical points in biomarker application is the choice of an end-point in preclinical or clinical trials for selected outcome measures, in order to achieve a significant change in selected parameters. Moreover, it is unfavorable for the treatment to affect biomarker readout in a way that does not correspond to disease regression/pathology, e.g., immune assay could be affected by organism response to the treatment procedure or reagent itself.

## 4. HD Wet Biomarkers

Wet biomarkers are analyzed from patient fluids, such as blood, saliva, urine and also CSF, in the case of invasive sampling. Blood is an especially attractive source of wet biomarker due to its easy and cost-effective collection, the minimally invasive nature of the procedure used, the large amount of material collected, as well as the wide range of analytical methods [48]. A single sample provides results for many interesting analytes that may correlate with the disease. Numerous molecules have already been proposed as wet biomarkers for HD (Table 2).

### 4.1. Neurofilament Light Chain

Among the most promising representatives of wet biomarkers is neurofilament light chain (NFL). It is the smallest subunit of neurofilaments and a major component of the neuronal cytoskeleton [49]. Concentration of NFL reflects neurodegeneration within the CNS, as it is released from damaged cells and indicates axonal degeneration. The levels of NFL protein in both CSF and the plasma are tightly correlated and reflect clinical severity in HD patients [49,50,51,52]. Interestingly, Byrne et al. indicated that NFL concentration was higher in plasma premanifest and early-stage HD patients than controls, even before the onset of symptoms. Their results showed that NFL level reflected baseline cognitive impairment and motor dysfunction depended on disease stage, age and the number of CAG repeat units. NFL is the first described biomarker from blood directly associated with a causative gene expansion [50]. The value of NFL as a dynamic and potent biomarker of brain atrophy has been highlighted in several studies.

Johnson et al. indicated a correlation between NFL concentration in plasma and cortical thinning and white matter volume reduction, assessed by MRI imaging, in both presymptomatic and symptomatic HD patients [52]. Significant progress in the development of analytical methods enables reliable measurements that largely reflect the status of the CNS. NFL can be determined using ultrasensitive techniques, such as immunomagnetic reduction (IMR) and the single molecule array (Simoa) in peripheral blood [53]. Results of the most recent research indicate that NFL concentration in blood is closely correlated with CSF and directly reflects neurodegeneration within the CNS in various neurodegenerative diseases, such as Parkinson’s disease (PD), Alzheimer’s disease (AD), frontotemporal dementia (FTD), amyotrophic lateral sclerosis (ALS), atypical parkinsonian disorders (APD), traumatic brain injury (TBI), Creutzfeldt–Jakob disease (CJD) or SCA3 [49,54,55,56,57,58,59]. In consequence, it may lead to becoming a reliable but not disease-specific biomarker of the risk and progression of neurodegeneration [49,50,57,60].

### 4.2. Mutant Huntingtin

The development of therapeutic strategies aimed at mutant huntingtin (mHTT) reduction indicates the importance of using this protein not only as a progression indicator but also as a measure for therapeutic effectiveness. The level of mHTT in CSF is significantly correlated with the severity of the disease [61], regardless of the age or the number of CAG repeats, and its measurement is possible using various assays. Due to the low endogenous level of HTT, ultra-sensitive assays are required and assays specific for mHTT are desired. Single molecule counting (SMC) immunoassay [61,62] as well as microbead-based immunoprecipitation and flow cytometry (IP-FCM) demonstrated high specificity for mHTT in CSF [63]. The mHTT level was also determined in the blood [64,65,66,67] and saliva [68,69]. These analyses unambiguously distinguish between HD patients and healthy individuals. The level of mHTT was significantly elevated in purified HD patient leukocytes as well as peripheral blood mononuclear cells (PBMCs) compared with controls, and it was correlated with disease progression [66,67]. Moreover, elevated mHTT levels in peripheral immune cells were noted in patients before the onset of symptoms and in the earlier phase of the disease, but no correlation in the mean mHTT level was found in relation to early and moderate phase HD patients [66]. Determining the level of the blood-soluble form of mHTT derived from the CNS is relatively difficult diagnostically due to the fact that its concentration is very low and constantly produced peripherally [70]. For this purpose, a highly sensitive time-resolved Förster resonance energy transfer (TR-FRET) [66] and homogeneous time-resolved Förster resonance energy transfer (HTRF) are used [64,65]. These assays are performed in peripheral whole blood, isolated erythrocytes or buffy coats containing majority of leukocytes and platelets after centrifugation. Another sensitive, relatively new method of assessing the level of both polyglutamine-independent human HTT and polyglutamine-expanded human HTT proteins in blood is the Meso Scale Discovery electrochemiluminescence immunoassay platform (MSD) [67,71]. Compared to the aforementioned TR-FRET immunoassay which requires purified leukocyte subpopulations, it can be performed in complex tissues and fluids, such as PBMCs. This makes it more suited to high-throughput studies. Due to the synergistic role of mHTT and NFL in the development of HD, the simultaneous use of both protein assays provides clear benefits in assessing progression and identifying potential therapeutic effects [51].

### 4.3. Oxidative Stress Markers

The background of HD pathophysiology is also being described by research concerning oxidative stress. Recent studies indicate that oxidative damage, along with mitochondrial dysfunction and impairment in the electron transport chain, plays a significant role in neurodegenerative processes [72,73]. However, despite a growing number of studies that indicate the importance of oxidative damage in HD, identification of the main pathways of disease pathogenesis associated with oxidative stress has not been fully explained. In HD, this effect is of particular importance in the progression of the disease in later stages, contributing to the worsening of the pathological condition [72]. Increased oxidative stress along with oxidative DNA damage has been reported in the peripheral blood of HD patients [74,75].

One measure of oxidative stress is uric acid (UA), the most abundant naturally occurring, strong antioxidant both in the CNS and in the periphery [76]. Evidence for the neuroprotective effect of uric acid has been provided by the results of studies, including clinical studies, for PD, indicating its potential as a biomarker of reduced risk of morbidity and milder progression of PD [77,78]. Corey-Bloom et al. very recently showed significantly lower levels of UA in the plasma and saliva of HD patients compared to healthy individuals. The same correlation was observed in an analysis of post mortem brains of HD patients in the mentioned study and other research [79]. Reduced antioxidant capacity in HD, as measured by UA levels, non-invasively in body fluids, makes this endogenous antioxidant another noteworthy, potentially significant indicator of disease symptoms.

A promising biomarker of oxidative stress in HD may also be the metabolites of the kynurenine pathway (KP) resulting from the degradation of tryptophan in the microglia. The released metabolites may be excitotoxic, and inhibition of kynurenine monooxygenase (KMO) showed therapeutic effects in HD [80,81]. In the blood, increased kynurenine to tryptophan ratio and decreased tryptophan have been reported in patients manifesting symptoms of HD. Such correlations were not found in premanifest patients, which may indicate an increased conversion of tryptophan to kynurenine in the later stages of HD [82,83]. KP metabolites seem to be interesting biomarkers; however, due to varied possibilities of crossing the BBB, they require confirmation in detailed and well-designed studies on a large number of samples from HD patients, both from CSF and blood [84].

8-hydroxy-2-deoxyguanosine (8-OHdG) seems to be another interesting, but uncertain, biomarker of oxidative stress in the pathogenesis of HD. As an indicator of oxidative DNA injury in HD patients, this analyte has been evaluated in many studies. Increased levels of 8-OHdG have been reported in blood, which have been associated with onset of the disease [75,85,86,87]. Unfortunately, the results of most studies on the participation of 8-OHdG in the pathogenesis of HD are contradictory [88,89], and the results of Borowsky et al. in 2013 suggested that it is not a suitable indicator of disease as a marker analyzed in blood [90].

### 4.4. BDNF

Brain-derived neurotrophic factor (BDNF), another potentially useful HD biomarker, is produced by cortical neurons. BDNF is necessary for the survival, growth and differentiation of nervous system cells [84,91]. By binding tropomyosin, the kinase B receptor (TrkB), it regulates the function of neurons both in the PNS and CNS [92]. Lowering BDNF levels in the brain due to disturbance of its transcription and mutant huntingtin-dependent expression plays a key role in the pathogenesis of HD. As a result of the mutation, huntingtin loses its beneficial activities, which leads to decrease in the production of BDNF in the cortex. BDNF transport from the cortex to the striatum is impaired and results in death of striatal neurons that have insufficient neurotrophic support [92]. Decreased levels of this factor have been reported in both post mortem studies of patient brains and in mouse HD models [93,94]. The results of studies determining the level of BDNF in the blood of patients, unfortunately, are very divergent (Table 2). Some authors report reduced levels of *BDNF* transcripts in whole blood [95], and BDNF protein in the serum of premanifest and manifest HD patients [40], also showing a correlation with the number of CAG repeats, motor and cognitive scores [96]. However, in other studies, no differences in serum or plasma BDNF levels were found between HD patients and healthy individuals [97,98]. To make this even more contradictory, one of the recent reports indicates increased levels of BDNF in the platelets of HD patients [99]. In addition, overexpression of BDNF in a HD mouse brain was shown to ameliorate the HD phenotype [100], and the BDNF knockout animals showed consistent symptoms like HD transgenic mice [101]. The disappointing conclusions of the use of BDNF as a blood biomarker for HD are also confirmed by the latest research by Gutierrez et al. [91], confirming previous reports indicating no changes in the concentrations of the tested protein in the plasma or serum of HD patients. Interestingly, the authors observed significantly lower levels of BDNF in saliva in premanifest and manifest HD patients in relation to the control, found in the 10 years prior to the onset of disease symptoms. This makes BDNF in the saliva a possible early, non-invasive biomarker of disease onset [91].

### 4.5. Metabolic Markers

Among the interesting molecules in terms of metabolic disorders associated with HD is 24 (S) hydroxycholesterol (24OHC), the major metabolite of cholesterol in the brain. Its decreased levels have been observed in the plasma of HD patients and correlated with a reduction in caudate volume [102,103]. This may prove that this metabolite reflects the progressive course of neuronal loss in HD [70]. Recent studies on this metabolite, carried out by Leoni et al., focused on relative plasma levels of 24OHC in three groups of gene-expanded individuals [103]. The authors indicate significant differences in the concentration of 24OHC in individual groups of HD patients; the mean value of the assessed metabolite decreased with the disease progression. The effect of lowering the 24OHC was greater than the changes in the cognitive and motor dysfunction or changes reported by brain imaging. In addition, the reduction in plasma 24OHC qualitatively reflects the progression of striatal atrophy. In the plasma of HD patients, decreased levels of the precursors of cholesterol, lanosterol and lathosterol and the bile acid precursor 27-hydroxycholesterol were also noted [103]. In order to further validate this biomarker, long-term observation of patients is suggested to determine the rate of change in metabolic markers during disease progression [70].

The plasma metabolome was studied by Rosas et al. using more than 100 HD early-symptomatic samples and control samples, as well as about 50 premanifest HD samples [104]. Specific metabolomic perturbations were described for HD and premanifest HD samples, including alterations in tryptophan, tyrosine and purine pathways.

Other metabolic disorders associated with the pathogenesis of HD have also been tested as potential biomarkers. In this case, the situation is similar—there are number of studies showing changes in the catabolic profile in premanifest and in the early stages of HD. Disturbed metabolism of amino acids, such as valine, leucine and isoleucine, in many studies was consistently correlated with weight loss, disease progression or the CAG repeats number [105,106,107]. Unfortunately, in other studies, changes in branched chain amino acids have not been reported [108,109,110]. Similar results were obtained for cholesterol; most analyses showed no changes in the peripheral levels of all its fractions [111,112,113], except one study, where a reduction in total cholesterol, HDL cholesterol and LDL cholesterol was found in both premanifest and manifest HD patients [97].

### 4.6. miRNAs

microRNAs (miRNAs) are conserved small non-coding RNAs that post-transcriptionally control gene expression, involved in regulating many physiological as well as pathological processes. According to current studies, mature miRNAs are not only present in cells but might also be sorted into exosomes, which are extracellular vesicles taking part in intercellular communication without direct cell-to-cell contact [114]. Such extracellular miRNAs, also called circulating miRNAs (cmiRNAs), are remarkably stable in extracellular environments, including body fluids, which makes them a potential candidate for diagnostic and prognostic biomarkers [115]. Abnormal miRNA expression patterns have been proposed to be related to etiology and progression of several polyQ diseases. For HD, numerous studies have indicated altered expression patterns of miRNAs in cell lines, tissues of mouse models and, importantly, patients’ brains (reviewed in [116]) (Table 2). Nevertheless, deregulation of miRNA expression in body fluids has been less intensively investigated.

The first report suggesting that microRNA changes observed in HD brains may be detectable in plasma was made by Hoss et al. in 2015, who identified two up-regulated miRNAs, miR-10b-5p and miR-486-5p, in both brain and blood samples from HD patients [117]. miR-10b-5p was shown to have a strong correlation with disease stage, age of onset and neuropathological changes in the brain [118]. Additionally, it was significantly elevated in symptomatic HD patients’ blood but not in asymptomatic HD gene carriers. One must know that the limitation of this study was the small number of samples in the presymptomatic (*n* = 4) and control (*n* = 8) groups [117]. Similar findings were reported by Chang et al. where they investigated the expression levels of 13 miRNAs in the peripheral leukocytes of 36 HD patients (together with 8 samples from asymptomatic HD carriers and 28 controls). Ten of those miRNAs had been previously reported to have different expression levels, but 3 were newly discovered as potential regulators of differentially expressed genes in brains of HD patients. Interestingly, only miR-9* in HD patient samples was significantly changed but not in the presymptomatic group [119], as shown for miR-10b-5p in a previously mentioned study [117]. This study additionally demonstrated that a potential candidate of miRNA biomarkers could also be found in blood cells. There seems to be only one report suggesting that miRNAs might be used as early diagnosis biomarkers for asymptomatic HD patients. Gaughwin et al. identified significantly up-regulated miR-34b in HD patients’ blood before the onset of clinical manifestations. Interestingly, the level of this miRNA is elevated in response to mHTT-Exon-1 overexpression in cell lines and is also regulated by apoptotic protein p53 [120]. There are also reported alternations in the expression profiles of 168 cmiRNAs (in 15 HD patients and 7 controls). The great majority of these plasma miRNAs was up-regulated in HD patients; four of them (miR-132-3p, miR-363-3p, miR-10b-5p and miR-486-5p) showed genome-wide changes in HD patients, both in brain and blood samples, but only 13 cmiRNAs were significantly changed [121].

cmiRNAs are easily detectable by sensitive and accurate qPCR methods. The procedure seems valid for plasma from small input volumes, and is not sensitive to RNases and freeze-thaw-induced RNA degradation [122]. This fact is necessary in an attempt to propose a peripheral biomarker for HD. In contrast, there are several downsides: poor representation of research studies, low reproducibility and often contradictory results. Differences among studies could be due to technical variability, such as starting material and isolation or RNA extraction method, but primarily, the normalization of data needs to be clarified and standardized.

### 4.7. Immune System

A growing body of evidence emphasizes the immune aspects of HD to play an important role in the progression and perhaps onset of the disease. Its specific immune cell phenotype could serve as an early prognostic marker [123]. A chronic state of inflammation is observed in the peripheral tissue of the diseased patients that did not yet have symptoms of HD. Circulation, concerning elevated levels of cytokines [124] and chemokines [125], in the plasma of HD patients correlated to disease progression can help researchers in the struggle against the disease. The phenotype of the HD mouse model, including impaired synaptogenesis, mobility and imbalanced blood cytokine levels, was improved after irradiation and transplantation of wild-type bone marrow cells [126]. Furthermore, irradiation-disrupted BBB showed an influx of monocytes into brain tissue and their differentiation towards microglia [127], although those cells are known to be resident cells of the brain [128] and are sustained by self-renewal [129]. Microglia are brain residual cells of an innate IS that originate from myeloid cells and are responsible for immune response towards environmental stress and tissue clearance [130]. Studies show that unstimulated microglia from HD patients already show pro-inflammatory activation status and are hyper-reactive upon endotoxin (LPS) stimulation, secreting IL-8 and TNFα [131,132]. All of the presented research indicates the long-known importance of IS in the pathology of HD. However, this fact has been known for more than 30 years, when the first evidence of the T cell responsiveness of HD patients with no change in the number of T cells was proven [133]. It is important to ask the following questions: what has been done with that knowledge in recent years? Can we conclude and proclaim any potential targets for strategies? Can we point out any immune biomarkers that could help to estimate the effectiveness of a therapeutic strategy or in early diagnosis, even if not a lone-standing but an additive biomarker of premanifest HD?

When isolated from peripheral blood, human HD macrophages showed an increase in phagocytosis compared to healthy controls [132]. A similar effect was found and further characterized on animal models of HD (R6/2, YAC128, Hdh150). Träger et al. showed that these models recapitulate a specific immune status of HD patients and can serve as good targets for the modulation of the disease or biomarkers. An increase in activated myeloid cells has been shown to be present in HD along with pronounced secretion of cytokines IL-1β, IL-6 and TNFα [132]. Those cytokines have been involved and positively correlated with dropping levels of wild-type HTT. It could be speculated that mHTT translated instead of wtHTT completely loses its function in immune and inflammatory contexts.

In addition to the abovementioned, TGFβ is another cytokine that could serve as a potential biomarker of the disease course [134]. Interestingly, gene expression levels of this molecule were neither found to be different in T lymphocytes nor granulocytes. Only monocytes of HD patients showed this phenomenon. Furthermore, TGFβ is known to be differently expressed in the course of HD and in differently polarized macrophages. When investigating premanifest and postsymptomatic HD patients, it has been found that macrophages shift their polarization status from M1 in pre-HD to M2 in symptomatic patients with the change in expression from IL-12 in presymptomatic and IL-10 in postsymptomatic patients. di Pardo et al. proved that the NFκB pathway is mainly involved in this conversion [134].

Among others, this discovery has been a milestone for clinical trials of Laquinimod in the treatment of HD patients. Dobson et al. showed that 5µM treatment of peripheral myeloid cells from HD patients “calms down” a hyper-reactive immune response towards bacterial endotoxin LPS [135]. It modulated IL-1β, IL-6, IL-8, IL-12p70 and TNFα mainly in manifest patients with some being changed in premanifest HD patients. Although the study considers that the shift is not due to the inhibition of the NFκB pathway and that the work of action of Laquinimod is under further investigation [135], it is still under consideration as an additive treatment for other strategies, as a LEGATO-HD clinical study failed to show an improvement in the motoric phenotype of patients.

Another type of immune biomarker that could be used in therapy efficiency assessment is the discovery made by Lee et al. [136]. This is the angiotensin 2 type 1 receptor autoantibody (AT1R-AA), which has been shown to be up-regulated in the serum of HD and MS patients. Over a hundred HD and MS patients were compared to over 100 healthy controls, and almost 38% of HD patients showed relevantly increased AT1R-AAs. Levels of AT1R-AAs have been positively correlated to disease burden. Patients with high levels of AT1R-AA had earlier disease onset [136]. This is not the only antibody found to be dysregulated in HD. A decade earlier, anti-gliadin antibodies (AGA) were found in 23 out of 52 HD patients taking part in the study [137].

### 4.8. Others

Melatonin was initially disqualified as a useful HD biomarker due to sample collection time-based relevance [138,139], but subsequent studies on a large cohort showed a significant decrease in its concentration in the blood with disease progression [140]. Proper melatonin secretion increases soon after dusk and in HD patients was significantly delayed compared to individuals without neurological disorders [138]. To properly assess circadian melatonin secretion, night sample plasma collection is essential; therefore, melatonin as a biomarker has limited usefulness. Similarly, contradictory results were obtained for growth hormone (GH), and some initial studies reported increased somatotropic activity associated with the severity of HD [71], while other studies do not confirm this relationship, showing no differences in GH levels in HD patients compared to controls [71,141,142].

## 5. Microbiome

Until scientific outbreaks in the last decade in the microbiome field, the placenta, uterus and brain had been thought to be sterile [143]. By post mortem analyses, it has been found that in normal healthy conditions, bacteria, in fact, colonize the abovementioned organs and specifically influence their function. Growing numbers of scientists devoted to understanding microbiota in different organs show us how important those microbes are for normal function of the organism. Furthermore, studies in many fields are now increasingly focused on the impact that the microbiome has during the course of disease. Implementing germ-free (GF) animals into research in many disorders has revealed developmental defects of normal functions due to the diminished microbiome. Interestingly, in GF mice, BBB is to some extent permeable. Braniste et al. showed lower levels of tight junction proteins of endothelium: occludin and claudin-5 in frontal cortex, striatum and hippocampus of adult brains [144]. Those structures are mostly impaired in the course of HD. Furthermore, it is more striking that those phenotypes could be reversed by colonization of the intestine with *Clostridium tyrobutyricum* and *Bacteroides thetaiotaomicron* that mainly produce butyrate and acetate and propionate, respectively [144]. In 2019, there was an attempt to identify the risk factor for HD in the brain microbiome. Researchers then showed three species of bacteria colonizing the human HD brain post mortem: *Pseudomonas*, *Acinetobacter* and *Burkholderia* [145]. The challenge is to prove how those bacteria enter the brain, where they come from, how they influence the course of HD and more importantly, how they are reflected in peripheral tissues (e.g., gut) and could serve as a biomarker of HD. Knowing that BBB can be “leaky”, allowing bacteria to enter and colonize the brain, could there be a possibility that the brain microbiome will escape and influence the gut microbiome?

Recent direct evidence that there are significant differences between healthy controls and HD patients with regard to microbiome diversity was proclaimed recently by Wasser et al. [146]. They were able to pinpoint lowered abundances of *Firmicutes*, *Lachnospiraceae* and *Akkermansiaceae* in fecal samples from HD patients. Interestingly, authors showed correlations between *E. halli* abundance and disease onset and progression. Still, authors are cautions about forming a strong conclusion given the small sample size (around 40 in each group), but the data they brought to light may have an impact on finding the peripheral biomarker in those bacterial species [146]. Correspondingly, in a mouse model of HD (R6/1), an increase in *Bacteriodetes* (Gram−), but a decrease in *Firmicutes* (Gram+), which corresponds to the abovementioned study on HD patients, was noticed [147]. Very similar results were achieved independently in another mouse model of HD (R6/2) with the addition of proven increased intestinal permeability [148].

How does it work? There are four proposed ways by which the microbiome can interact and have an impact on the development and course of neurodegenerative diseases [149]. First, defining the mucosal immune system (e.g., Th17 response) and its homeostasis microbiome dysbiosis can lead to autoimmune response. Second, a simple change in microflora can leave room for other microbes and, as a consequence, infection and inflammatory response. The third way includes *nervus vagus* and its stimulation and an interaction between short-chain fatty acids (SCFA) and amyloid that can induce neuronal degeneration. The last proposed mode of action is through an influence of SCFA on maturation of microglia in the CNS, which impacts neuroimmune modulation and can trigger neurodegeneration of neurons [149,150].

**Table 2 ijms-22-01561-t002:** Proposed biomarkers found to be dysregulated in periphery of HD patients.

Blood Biomarkers	Type of Deregulation	References
NFL	up-regulation	[50,51,52]
mHTT	up-regulation	[64,65,66,67]
8-OHdG	up-regulation no difference	[75,85,86,87] [86,90]
UA	up-regulation	[76]
Kynurenine	no difference up-regulation	[82] [83]
Tryptophan	down-regulation no difference	[82] [83,104]
Kynurenine/tryptophan ratio	up-regulation	[82,83]
BDNF	down-regulation (transcript only)	[95]
	down-regulation	[40,96]
	no difference	[91,97,98]
	up-regulation	[99]
24OHC	down-regulation	[102]
Melatonin	no difference down-regulation	[138] [140]
GH	no difference up-regulation	[141,142] [141,151]
Metabolism of amino acids (valine, leucine and isoleucine)	changed	[107,108,109]
	not changed	[108,110,111]
Cholesterol profile	no difference down-regulation	[111,112,113] [97]
AT1R-AA	up-regulation	[136]
AGA	up-regulation	[137]
Serotonin N-acetylserotonin	down-regulation	[104]
	up-regulation	[104]
**Cytokines**		
Eotaxin, MCP-1, MCP-4, MIP-1β	up-regulation	[125]
IL-4, IL-8, IL-10, TNFα, GM-CSF	up-regulation	[124]
MMP-9, VEGF, IL-18	up-regulation	[152]
TGF-β1	down-regulation	[134]
IL-6	up-regulation	[132]
**miRNAs**		
miR-34b	up-regulation	[120]
miR-10b-5p, miR-486-5p	up-regulation	[117]
miR-30d-5p, miR-877-5p, miR-425-5p, miR-223-3p, miR-223-5p, miR-222-3p, miR-338-3p, miR-130b-3p, miR-628-3p, miR-361-5p, miR-128, miR-22-5p, miR-942	up-regulation	[121]
miR-9	down-regulation	[119]

## 6. Conclusions and Perspectives

In recent years, studies of AD, PD and ALS have revealed pathological pathways present in other tissues. Due to some similarities in the peripheral effects in various neurodegenerative disorders, it is worth considering biomarkers which can be used for each of these diseases [153]. Nevertheless, specific diseases, such as HD, are also characterized by some specific features, which can be analyzed as unique HD biomarkers.

Numerous and potentially reliable biomarkers involved in the pathogenesis and development of HD often present disappointing results. The following question remains: why are they low-repeat and often contradictory? The inconsistency of the results obtained for these analities in individual studies could be partially explained by the differences resulting from the tests used, the method of collecting, preparing or storing samples, as well as the stability of the tested analyte. Moreover, most of them, apart from the CNS, are also produced peripherally. A suitable example is the very promising BDNF, released by megakaryocytes and platelets, which may have a significant impact on the obtained results [84]. Additionally, some of them, such as 8-OHdG, do not show specificity for HD. Its level is also significantly increased in the course of other diseases, including cardiovascular pathology [154,155]. This is a common problem when determining potential biomarkers closely related to a specific disease entity, which complicates the interpretation of results and affects their credibility. Among the most important aspects in the research of peripheral biomarkers in biofluids seems to be the cohort size and its detailed characteristics. Research on a small population of patients, incomplete evaluation (in terms of detailed research at various levels), methodological inconsistency, lack of exclusion of confounding factors, inadequate assignment to research groups or incorrect control group make the results difficult to compare with each other, and the final effect is their large discrepancy. An important aspect of the biomarker’s usefulness for neurodegenerative diseases is its clinical utility, reproducibility and the possibility of its use in the assessment of the effectiveness of therapy. To establish such a marker, all conditions should be met. Currently, it is difficult to determine such a biomarker among the known and described ones. The most noteworthy and useful biomarkers seem to be mHTT in CSF for the evaluation of HTT reduction in clinical trials, as well as NFL, which has a prognostic value for HD and other NDDs. Furthermore, its assessment is minimally invasive.

There is no doubt that for complete success in therapy for HD, advances must come from both sides: (I) development of the most effective, specific and least invasive approach and (II) elaboration of a set of reliable biomarkers (Figure 1). Disease-modifying therapeutic approaches for HD are relatively highly advanced. Numerous mutant protein-targeting strategies have also been tested, as well as therapies aimed at initially disrupted pathways in HD [156,157]. The most direct approach of genetic disease therapy is to “repair” the mutation at the DNA level. In recent years, a new generation of tools for genome editing, such as ZFNs (zinc finger nucleases), TALENs (transcription activator-like effector nucleases) and CRISPR/Cas9, has generated a great interest and made technological progress [158]. Nevertheless, their clinical application is far less advanced than the use of RNAi tools and ASOs. The clinically tested HD-modifying approach using ASOs is hoped to be accessible for patients in the near future. As for now, this kind of treatment requires large doses of therapeutic synthetic reagent, and allele-selective targeting of *HTT* may be required for long-term intervention, instead of the non-allele-selective approach applied. An increasing number of studies aim to make treatments minimally invasive, e.g., with the use of AVVs that can be administered intravenously and cross the BBB and transduce neurons in the CNS [159].

Microbiome research in neurodegenerative diseases is an emerging field. It has been shown to have a great impact on many diseases, including cardiovascular disorders, such as hypertension, and that modulating it may bring beneficial effects [160]. Moreover, a clear connection between the microbiome and immune system is stated. Although recent studies mostly apply model organisms and some investigate patients, there remain vast unknown fields. We may not be able to modify the disease by just applying a new bacteria cocktail, but one could imagine an additive stimulation via the gastrointestinal tract to improve the everyday life of HD patients.

## Figures and Tables

**Figure 1 ijms-22-01561-f001:**
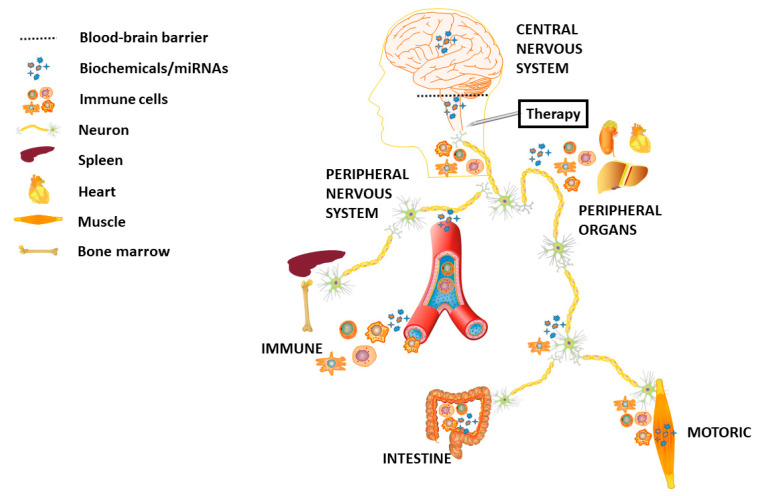
Conceptual visualization of therapeutic approach with an impact across a human body. In the hunt for HD biomarkers able to measure an outcome of treatment, one might choose from a whole spectrum of analytes ranging from CNS, through PNS, peripheral organs and blood, ending up in motoric skills of HD patients.

**Table 1 ijms-22-01561-t001:** Pharmaceutical company-tested therapeutic strategies for Huntington’s disease (HD), using molecules that target huntingtin mRNA that are in clinical testing or close to the start of this phase.

Company	Technology/Strategy	Molecule	Clinical Phase
Ionis Pharmaceuticals/Roche	ASO/Non-allele-selective	RG6042	I/II—completed III—scheduled for completion in 2023
Wave Life Sciences/Takeda	ASO/Allele-selective	WVE-120101, WVE-120102	I/II—scheduled for completion in 2020
UniQure Biopharma	RNAi/Non-allele-selective	rAAV5-miHTT (AMT-130)	I/II—scheduled for completion in 2022
Voyager/Sanofi-Genzyme	RNAi/Non-allele-selective	AAV-shRNA (VY-HTT01)	-
Spark Therapeutics	RNAi/Non-allele-selective	AAV-shRNA	-

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
