# Peer review of "What, When and How to Measure—Peripheral Biomarkers in Therapy of Huntington’s Disease"

_ijms, 2021, doi:10.3390/ijms22041561_

Round 1

Reviewer 1 Report

This manuscript is a very extensive review covering a vast array of biomarkers that have been measured in peripheral fluids of patients with Huntington’s disease (HD). Overall biomarkers in HD are needed for several reasons, including estimating disease onset, monitoring disease progression, gauging symptoms severity or outcomes and tracking therapeutics.  This idea should be presented early, since not all biomarkers address each of these areas.  (For example, in the Abstract “to assess the efficiency of the treatment”; in the Intro “…the goal is ….to be used as prognostic biomarkers”; in Section 3, “An appropriate biomarker should….correlates with disease progression”).  In addition, there are several areas of improvement in the writing, including rearrangement of a few paragraphs that could help the flow of the manuscript. Specific comments for improvement are provided below.

  1. The paragraph in the introduction on cardiac dysfunction in HD (starting at line 59) can be shortened to one or two sentences on peripheral dysfunction, unless the authors want to include equivalent discussions on dysfunction of other peripheral organs, i.e. skeletal muscle.
  2. Online 119, add the year that the gene therapy was approved by the FDA.
  3. For heading #2, “Disease-modifying strategies for HD”, only gene targeting strategies are discussed. Other types of therapies should be mentioned.  (See Dash & Mestre, Neurotherapeutics (2020) for other disease modifying treatment approaches).  Also, I would suggest moving the “Disease-modifying strategies for HD” to after section 5. Addressing ways to identify disease before ways to treat disease might improve the overall flow of the manuscript.
  4. Pg 4. How much lowering of total Htt is tolerated in mice?  50%?
  5. Section 3 provides a quick overview of the types of biomarkers, including neuroimaging in HD. However, the rest of the manuscript focuses solely on wet biomarkers (with a few neuroimaging studies correlated with wet biomarkers).  This section should be reworked or another section added to include the review of neuroimaging studies.
  6. It would be helpful if specific references were added for statements made in lines 193-195.
  7. Lines 224-225, or the paragraph that follows, are confusing as written. Biomarkers have been used in pre-clinical HD studies, and current clinical trials also use several biomarkers to monitor the effects of their treatment on disease parameters.
  8. Section 4, line 241: blood collection is not non-invasive, but rather is minimally-invasive.  Saliva collection is non-invasive if one wants to make the distinction.
  9. Lines 291/292 are confusing. I think the authors mean that highly sensitive methods, such as TR-FRET have been developed for detection in fluids when low levels of proteins are expected.
  10. With regards to melatonin as a biomarker, it could be worth mentioning that the timing of sample collection is critical (must be taken at the onset of darkness), so that could limit it’s usefulness.

Reviewer 2 Report

This review describes studies to provide therapeutic targets for HD. The authors focused on mining HD biomarkers to predict disease symptomatic onset or monitor disease progression using peripheral materials. Mainly, this introduces ‘wet biomarker’ such as blood, saliva, urine, and CSF as useful biomarkers and provided the importance and simplicity to use. However, peripheral effects are similar in various neurodegenerative diseases. As in that point, this manuscript also showed the specific features of HD and suggested the possibility of being analyzed as selective and specific HD biomarkers. 

  1. Overall, the manuscript provides a clear focus and appropriate approaches to its direction. The approaching point of this review is enough to be new and useful. 
  2. This work is helpful to overview the trials in experimental and clinical studies for HD. This review is well organized, introduced the very recent reports, and provides some controversial results.
